# Number of Bacteria in Saliva in the Perioperative Period and Factors Associated with Increased Numbers

**DOI:** 10.3390/ijerph19137552

**Published:** 2022-06-21

**Authors:** Yuki Sakamoto, Arisa Tanabe, Makiko Moriyama, Yoshihiko Otsuka, Madoka Funahara, Sakiko Soutome, Masahiro Umeda, Yuka Kojima

**Affiliations:** 1Department of Dentistry and Oral Surgery, Kansai Medical University Medical Center, Osaka 5708507, Japan; tanabear@takii.kmu.ac.jp (A.T.); moriyamm@takii.kmu.ac.jp (M.M.); 2Park Dental Clinic, Kanagawa 2510025, Japan; new.boss.yoshi@gmail.com; 3School of Oral Health Sciences, Kyusyu Dental University, Fukuoka 8038580, Japan; macloca02@gmail.com; 4Department of Oral Health, Nagasaki University Graduate School of Biomedical Sciences, Nagasaki 8528102, Japan; sakiko@nagasaki-u.ac.jp; 5Department of Clinical Oral Oncology, Nagasaki University Graduate School of Biomedical, Sciences, Nagasaki 8528102, Japan; mumeda@nagasaki-u.ac.jp; 6Department of Dentistry and Oral Surgery, Kansai Medical University Hospital, Osaka 5731191, Japan; kojimayk@hirakata.kmu.ac.jp

**Keywords:** bacteria in saliva, infection, perioperative oral management, surgery

## Abstract

Perioperative oral management is performed to prevent postoperative complications, but its indication and management method are unclear. This study aimed to investigate salivary bacterial counts pre-and postoperatively, and factors related to increased bacterial count postoperatively. We included 121 patients who underwent surgery under general anesthesia and perioperative oral management. The bacterial count in saliva was determined preoperatively, and first and seventh days postoperatively using the dielectrophoresis and impedance measurement methods. The relationships between salivary bacterial count and various variables were analyzed using one-way analysis of variance, Spearman’s rank correlation coefficient, and multiple regression analysis. The salivary bacterial count increased significantly on the first day postoperatively but decreased on the seventh day. Multivariate analysis showed that age (*p* = 0.004, standardized coefficient β = 0.283) and xerostomia (*p* = 0.034, standardized coefficient β = 0.192) were significantly correlated with increased salivary bacterial count preoperatively. Salivary bacterial count on the day after surgery was significantly increased in patients with a large number of bacterial counts on the day before surgery after preoperative oral care (*p* = 0.007, standardized coefficient β = 0.241) and postoperative fasting (*p* = 0.001, standardized coefficient β = −0.329). Establishing good oral hygiene before surgery and decreasing salivary bacterial count are necessary in patients with a high risk of postoperative surgical site infection or pneumonia, especially in older adults or postoperative fasting.

## 1. Introduction

Postoperative complications may occur after invasive surgical procedures such as cancer surgery, cardiac surgery, and organ transplant surgery. Some complications may be caused by pathogenic microorganisms derived from the oral cavity, and to reduce the risk of such complications, a clean oral environment and elimination of the infectious source in the oral cavity are essential [1,2,3,4]. In Japan, oral care before and after surgery has been included in the public medical insurance since 2012 and is commonly performed. In addition, the risk of surgical site infection (SSI) and postoperative pneumonia is significantly reduced by perioperative oral care during surgeries, especially during esophageal cancer [5,6,7,8,9,10], gastric cancer [10], colon cancer [10,11,12], lung cancer [13,14], liver cancer [15], and oral cancer [16], and procedures involving the cardiac [17] and brain [18]. Perioperative oral care has two primary purposes. The first is to treat or extract teeth that are potential sources of infection, such as teeth with periapical lesions and/or periodontal disease, thereby reducing the risk of developing SSI in areas distant from the oral cavity, such as cardiac prosthetic valves and hip and knee joint prostheses. Such SSI is believed to be transmitted through a blood-borne route from oral infectious sites. The second is to reduce the number of bacteria in saliva, thereby decreasing the risk of SSI in the head and neck, and upper gastrointestinal tract cancer surgery due to direct exposure to saliva and the risk of postoperative pneumonia due to aspiration of saliva.

However, the indications and effective methods of perioperative oral care have not been established. We focused on the number of bacteria in saliva, as the reduction in the number of bacteria in saliva is an objective of perioperative oral care. Some researchers have described that dental plaque is a reservoir of salivary bacteria and that plaque removal is important in preventing aspiration pneumonia [19,20]. However, it was reported that completely plaque-free, edentulous patients develop surgical site infections as often as those with teeth after cancer surgery [21], and elderly people requiring long-term care with edentulous jaws develop aspiration pneumonia as often as those with teeth [22]. These facts indicate that dental plaque is not the main reservoir of bacteria in saliva. The surface of the dorsum of the tongue may be one of the reservoirs of oral bacteria [23], but the reservoir of microorganisms in the saliva remains unclear. This study aimed to determine patients in which the number of bacteria in saliva increases after surgery and patients and diseases for which perioperative oral care is indicated.

## 2. Materials and Methods

### 2.1. Study Design and Patients

The participants were patients who underwent surgery under general anesthesia at Kansai Medical University Medical Center between 1 March 2021 and 31 December 2021, and who received perioperative oral care at the Department of Dentistry and Oral Surgery of the Center before, and on the first and seventh days after the surgery. The inclusion criteria were that the patient was at least 20 years of age and had a preoperative panoramic X-ray examination. The exclusion criterion was that the patient was discharged within 7 days of surgery. Since this study is an exploratory observational study, the sample size was not calculated statistically, and the number of patients during the study period was used as the sample size.

### 2.2. Variables

The following variables were examined: sex, age, primary disease, body mass index, smoking habit, drinking habit, routine dental management at a family dental clinic, serum total protein and albumin level before surgery, O’Leary plaque control record (PCR) [24], presence of periodontal pocket, number of residual teeth, denture use, oral wetness, operation time, oral intake condition on the day after surgery, and number of bacteria in saliva before and after surgery. Oral wetness was quantified according to the classification by Kakinoki et al. [25]. Saliva was collected by immersing a cotton swab in the posterior floor of the mouth. The number of bacteria in saliva was determined with a rapid bacteria quantification system (Panasonic Healthcare Co. Ltd., Osaka, Japan) using dielectrophoresis and impedance measurement methods.

### 2.3. Statistical Analysis

All statistical analyses were performed using SPSS ver. 26.0 (Japan IBM Co., Ltd., Tokyo, Japan). The relation between each variable and the number of bacteria in saliva was analyzed. One-way analysis of variance was used for analyzing categorical variables, and Spearman’s rank correlation coefficient was used for analyzing continuous variables. Multivariate analysis was performed using multiple regression analysis including factors found to be significant in univariate analysis and some other factors of interest. A two-tailed *p*-value < 0.05 was defined statistically significant.

## 3. Results

### 3.1. Patient Characteristics

Of the 147 patients registered, 121 were ultimately enrolled in the study, excluding 11 patients who were discharged within 7 days, 4 patients whose surgery was postponed, and 1 patient who was transferred to another hospital.

Fifty-four patients were males and 67 were females, with an average age of 69.4 years. The primary diseases were bone/joint disease in 41 patients, and gastrointestinal disease in 25 patients, followed by liver/gallbladder/pancreatic disease, urinary tract disease, cardiovascular disease, and lung disease. The average number of remaining teeth was 20.5, and 12 patients had undergone tooth extractions between the first visit and surgery. Of the 121 patients, 51 had xerostomia, of which 14 had moderate or severe xerostomia. The average operation time was 214 min. Forty-seven patients were fasting on the next day after surgery, and seven patients were fasting on the seventh day postoperatively (Table 1).

### 3.2. Changes in the Number of Bacteria in Saliva before and after Surgery

The average base 10 logarithm of the number of bacteria in saliva before oral care the day before surgery was 5.38. After oral care before surgery, the number of bacteria in saliva decreased to approximately 1/10 of the original count. On the day after surgery, the number of bacteria in saliva increased markedly to a value of 5.68. Thereafter, the number of bacteria in saliva decreased, and the value on the seventh day after surgery was restored to that before surgery (Figure 1).

### 3.3. Factors Related to the Number of Bacteria in Saliva before Surgery

Univariate analysis of factors associated with the number of bacteria in saliva before oral care on the day before surgery showed that older adults (*p* = 0.005) and xerostomia (*p* = 0.004) were significantly associated with increased number of bacteria in saliva (Table 2). Multiple regression analysis was performed with five factors entered as covariates: age and xerostomia, which were found to be significant in univariate analysis, and smoking, PCR, and regular dental checkups, which were believed to potentially affect the number of bacteria in saliva. The results showed that older age (*p* = 0.004, standardized coefficient β = 0.283) and xerostomia (*p* = 0.034, standardized coefficient β = 0.192) were significantly associated with the preoperative number of bacteria in saliva (Table 3).

### 3.4. Factors Related to the Number of Bacteria in Saliva on the Day after Surgery

Five factors were found to influence the increase in the number of bacteria in saliva the day after surgery in univariate analysis: older adults (*p* = 0.016), smoking (*p* = 0.040), low albumin level (*p* = 0.035), increased number of bacteria in saliva after oral care the day before surgery (*p* = 0.005), and postoperative fasting (*p* < 0.001) (Table 4). Multiple regression analysis including these five univariate significant factors with covariates revealed that an increased number of bacteria in saliva after oral care the day before surgery (*p* = 0.007, standardized coefficient β = 0.241) and postoperative fasting (*p* = 0.001, standardized coefficient β = −0.329) were significantly associated with an increased number of bacteria in saliva on the day after surgery (Table 5).

## 4. Discussion

The study results showed that the number of bacteria in saliva after surgery was significantly increased in patients with an increased number of bacteria in saliva after oral care the day before surgery and in those who were fasting postoperatively.

Perioperative oral care significantly reduces the risk of developing SSI and postoperative pneumonia after various surgeries. The goals of preoperative oral management, as mentioned above, are to eliminate the sources of oral infection and suppress an increase in the number of bacteria in saliva. Therefore, preoperative oral management in the form of providing oral hygiene instructions, dental calculus removal, tooth polishing, and extraction of teeth that may become a source of infection, is performed to establish a good oral environment.

Direct exposure of wounds such as those of head and neck and esophageal cancers to saliva containing pathogenic microorganisms can result in SSI. Funahara observed that the number of bacteria in saliva remarkably increased after surgery in patients with oral cancer who had undergone flap reconstruction and tracheal incision [26]. However, local application of an antibacterial ointment in the oral cavity decreased the number of bacteria for a certain period. Postoperative SSI can be significantly suppressed by topical application of an antibacterial drug in the oral cavity 48 h after surgery [27]. However, unintentional aspiration of saliva in the lower respiratory tract, especially in cases with compromised immunity, can cause aspiration pneumonia. Soutome and Iwata reported that pre- and postoperative oral management significantly suppressed the incidence of postoperative pneumonia in patients after esophageal and lung cancer surgeries [2,11]. The purpose of this study was to clarify whether the number of bacteria in saliva increases after surgery and the patients in whom such an increase may occur. Funahara determined the number of bacteria on the dorsum of the tongue before and after surgery in 54 cases of cancer or heart surgery and reported that the number of bacteria increased in patients with high dental plaque scores and no postoperative oral feeding [28]. However, as far as we know, factors that affect the number of bacteria in saliva in the perioperative period have not been reported.

If factors related to the increase in the number of bacteria in saliva in the postoperative period are clarified, several complications can be prevented by focusing on the oral care of patients with such factors. The results of this study showed that the number of bacteria in saliva before preoperative oral care was higher in elderly patients and those with xerostomia. This could be attributed to the decrease in salivary secretion, oral functions such as tongue pressure, and masticatory power with age [29]. As a result, the self-cleaning effect in the oral cavity decreases. We have reported that swallowing function decreases in elderly patients requiring long-term care with decreased tongue pressure, and decreased tongue pressure is associated with an increase in the number of bacteria in saliva [30]. Although we did not investigate tongue pressure and masticatory power in this study, the self-cleaning effect could be reduced and the number of bacteria in saliva could be increased in elderly patients because of decreased oral function. Further, the number of bacteria in saliva increased significantly on the day after surgery, especially in those with a greater number of bacteria after oral care on the day before surgery and who were fasting postoperatively. A decrease in the number of bacteria in saliva by preoperative oral care led to a decreased number of bacteria in saliva the day after surgery. Therefore, preoperative oral care is indicated in surgical patients at high risk of SSI and postoperative infection. Adequate oral care and instructions regarding the use of antimicrobial mouthwashes are particularly important. Patients who do not eat orally the day after surgery may have compromised oral self-cleaning. Adjuvant measures such as frequent gargling are required for such patients.

The limitations of the study should be considered. First, the number of cases is small as it is a single-center study, therefore, the results cannot be generalized. Next, the number of bacteria in saliva was determined using a bacterial counter, and the bacterial species were not investigated. Furthermore, the endpoint was the number of bacteria in saliva, which is not a postoperative complication. The number of cases was small and the frequency of postoperative complications was low. A large number of cases is required to consider postoperative complications as the endpoint. We aim to examine the relationship of the number and species of bacteria in saliva with the occurrence of postoperative complications with a greater number of cases and using real-time polymerase chain reaction (PCR) in the future.

## 5. Conclusions

The number of bacteria in saliva increases markedly after surgery, especially in those with a greater number of bacteria after oral care on the day before surgery and who do not eat orally on the day after surgery. Establishing good oral hygiene before surgery and a method to suppress the increase in the number of bacteria in saliva of patients with a high risk of postoperative SSI or postoperative pneumonia is essential, especially in elderly patients and when postoperative oral feeding is not possible.

## Figures and Tables

**Figure 1 ijerph-19-07552-f001:**
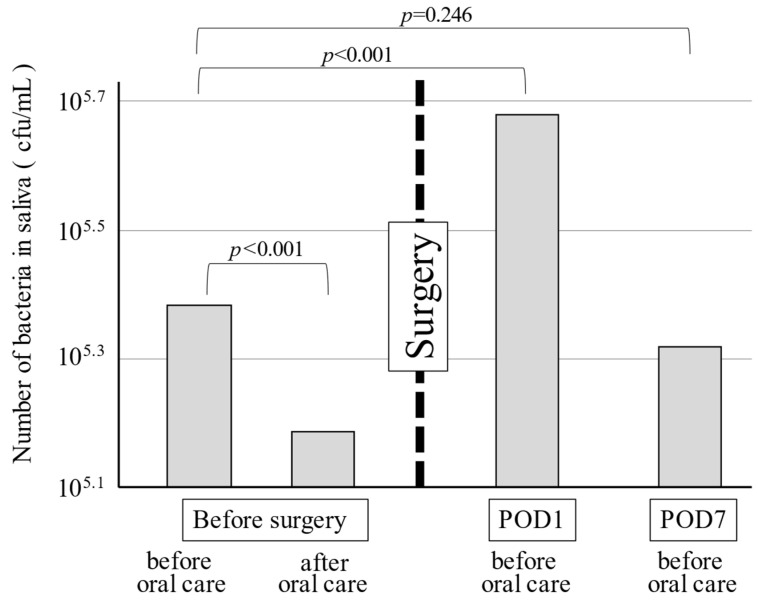
The number of bacteria in saliva during the perioperative period.

**Table 1 ijerph-19-07552-t001:** Patients’ characteristics.

Variable		Number of Patients/ Mean ± Standard Deviation
Sex	Male	54
	Female	67
Age		69.4 ± 12.3
Primary disease	bone/joint	41
	colon	15
	uterus/ovaries	15
	cardiovascular	10
	liver	7
	pancreas	6
	lung	6
	esophagus	5
	gastric	5
	gall bladder	3
	others	8
Body mass index		22.8 ± 3.70
Smoking habit	(−)	93
	(+)	28
Drinking habit	(−)	96
	(+)	25
Regular dental management	(−)	61
	(+)	60
Serum total protein (preoperative)		6.99 ± 0.547
Serum albumin (preoperative)		4.01 ± 0.530
Plaque control record (preoperative)		56.9 ± 27.2
Periodontal pocket	<6 mm	115
	≥6 mm	6
Tooth extraction before surgery	(−)	109
	(+)	12
Number of teeth		20.5 ± 8.74
Denture use	(−)	80
	(+)	41
Dry mouth (preoperative)	grade 0	70
	grade 1	37
	grade 2	11
	grade 3	3
Operation time (minutes)		214 ± 138
Feeding condition (next day after surgery)	Fasting	47
	Orally	74
Feeding condition (7th day after surgery)	Fasting	7
	Orally	114

**Table 2 ijerph-19-07552-t002:** Factors related to the number of bacteria in saliva before surgery (univariate analysis).

Variable		Mean ± Standard Deviation	Correlation Coefficient	*p*-Value
(i) Category data				
Sex	male	5.40 ± 0.472		0.763
	female	5.37 ± 0.535		
Smoking habit	(−)	5.39 ± 0.516		0.891
	(+)	5.37 ± 0.478		
Drinking habit	(−)	5.42 ± 0.537		0.126
	(+)	5.24 ± 0.334		
Regular dental management	(−)	5.37 ± 0.518		0.784
	(+)	5.40 ± 0.498		
Denture use	(−)	5.36 ± 0.475		0.419
	(+)	5.43 ± 0.565		
Periodontal pocket	<6 mm	5.38 ± 0.507		0.491
	≥6 mm	5.52 ± 0.510		
Extraction before surgery	(−)	5.38 ± 0.516		0.959
	(+)	5.38 ± 0.421		
Dry mouth	grade 0–1	5.32 ± 0.438		* 0.005
	grade 2–3	5.64 ± 0.681		
(ii) Continuous data				
Age			0.263	* 0.004
Body mass index			0.041	0.656
Serum total protein			−0.064	0.490
Serum albumin			−0.062	0.518
Number of teeth			−0.063	0.493
Plaque control record			0.079	0.391
Number of bacteria on the tongue			0.058	0.527

* indicates a significant difference.

**Table 3 ijerph-19-07552-t003:** Factors related to number of bacteria in saliva before surgery (multivariate analysis).

Variable	Unstandardized Coefficient		Standardized Coefficient	95% Confidence Interval of B		*p*-Value
	B	SE	β	Lower	Upper	
Age	0.012	0.004	0.283	0.004	0.019	* 0.004
Smoking habit	0.076	0.108	0.063	−0.139	0.290	0.485
Plaque control record	0.001	0.002	0.032	−0.003	0.004	0.732
Routine dental management	0.076	0.090	0.076	−0.102	0.254	0.399
Dry mouth (preoperative)	0.246	0.114	0.192	0.020	0.473	* 0.034

* indicates a significant difference.

**Table 4 ijerph-19-07552-t004:** Factors related to the number of bacteria in saliva on the day after surgery (univariate analysis).

Variable		Mean ± SD	Correlation Coefficient	*p*-Value
(i) Category data				
Sex	male	5.73 ± 0.667		0.528
	female	5.66 ± 0.514		
Smoking habit	(−)	5.76 ± 0.633		* 0.040
	(+)	5.48 ± 0.612		
Drinking habit	(−)	5.74 ± 0.654		0.135
	(+)	5.51 ± 0.539		
Regular dental management	(−)	5.74 ± 0.691		0.436
	(+)	5.64 ± 0.574		
Denture use	(−)	5.66 ± 0.643		0.427
	(+)	5.76 ± 0.626		
Periodontal pocket	<6 mm	5.71 ± 0.642		0.231
	≥6 mm	5.34 ± 0.422		
Tooth extraction before surgery	(−)	5.68 ± 0.632		0.503
	(+)	5.81 ± 0.691		
Dry mouth (preoperative)	grade 0–1	5.62 ± 0.618		0.064
	grade 2–3	5.90 ± 0.676		
Feeding condition (next day after surgery)	fasting	6.00 ± 0.702		* < 0.001
	orally	5.48 ± 0.495		
(ii) Continuous data				
Age			0.222	* 0.016
Body mass index			0.036	0.702
Serum total protein (preoperative)			−0.098	0.293
Serum albumin (preoperative)			−0.202	* 0.035
Number of teeth			−0.098	0.294
Number of bacteria in saliva (preoperative, before oral care)			0.173	0.057
Number of bacteria in saliva (preoperative, after oral care)			0.254	* 0.005
Plaque control record (preoperative)			0.126	0.175
Number of bacteria on the tongue (next day after surgery)			0.133	0.154
Operation time (minutes)			0.131	0.161

* indicates a significant difference.

**Table 5 ijerph-19-07552-t005:** Factors related to the number of bacteria in saliva on the day after surgery (multivariate analysis).

Variable	Unstandardized Coefficient		Standardized Coefficient	95% Confidence Interval		*p*-Value
	B	SE	Β	Lower	Upper	
Age	−0.002	0.005	−0.035	−0.011	0.008	0.701
Smoking habit	−0.197	0.130	−0.132	−0.456	0.061	0.133
Serum albumin	−0.107	0.117	−0.087	−0.339	0.124	0.360
Dry mouth (preoperative)	0.158	0.143	0.097	−0.126	0.442	0.272
Number of bacteria in saliva (preoperative, after oral care)	0.456	0.166	0.241	0.126	0.786	* 0.007
Feeding condition (next day after surgery)	−0.432	0.121	−0.329	−0.673	−0.191	* 0.001

* indicates a significant difference.

## Data Availability

The datasets used and analyzed during the study are available from the corresponding author upon reasonable request.

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
