# Peer review of "Number of Bacteria in Saliva in the Perioperative Period and Factors Associated with Increased Numbers"

_ijerph, 2022, doi:10.3390/ijerph19137552_

Round 1

Reviewer 1 Report

The manuscript submitted by Yuki Sakamoto et al, reported the importance of perioperative oral care before the invasive surgical procedures and investigated the number of bacteria in saliva before and after the surgical procedure.

The topic of preoperative oral management is a critical issue for both clinicians and patients and this research will help them predict the decreased risk of postoperative infection.

The manuscript itself is well constructed and clear.

The English is clear and spelled correctly.

The introduction is short and insufficient. I suggest you to add more information.

I do not have any comments regarding the methodology and significance of this important issue covered in this paper.

The discussion is well-conducted, supported by the results of study and research from the literature.

The conclusion are correctly.

Author Response

The introduction is short and insufficient. I suggest you to add more information.

(Reply)

Line 54-62: Some sentences were added as follows. 

Some researchers described that dental plaque is a reservoir of salivary bacteria and that plaque removal is important in preventing aspiration pneumonia [16, 17]. However, it was reported that completely plaque-free, edentulous patients develop surgical site infections as often as those with teeth after cancer surgery [18], and elderly people re-quiring long-term care with edentulous jaws develop aspiration pneumonia as often as those with teeth [19]. These facts indicate that dental plaque is not the main reservoir of bacteria in saliva. The surface of the dorsum of the tongue may be one of the reservoirs of oral bacteria [20], but the reservoir of microorganisms in the saliva remains unclear.

Reviewer 2 Report

The 121 patients who underwent surgery under general anesthesia and perioperative oral management. This article shows the large number of bacterial in saliva were detected in before and after surgery. This manuscript contains important information, however, this version did not achieve a sufficient contents for IJERPH. Please provide more information for patients characteristics. The authors should describe detail for Gastrointestinal tract diseases. Such as, number of gastric surgery, esophagus surgery, colon surgery should be provide. In addition, categorized into liver, bladder and pancreas is not suitable for this study, because the quality of each operation varies widely and simple comparison/categorization of them is not possible. Please add these information in Table 1.

In Figure1 shows the large number of bacteria was detected in POD1 group  . In contrast,  the small number of bacteria was detected in POD7 group. Please let us know whether these groups are fasting or orally. The authors described in this article, eat orally is very important to suppress the increase in the number of bacteria in saliva of patients. The authors should add more information about fasting or orally in postoperative period. 

In page 7, in line 170, the authors said, the number of bacteria is higher in saliva before operation in elderly patients. This could be attributed to the decrease in salivary secretion. It should need some citation literature for straightforward relationship with decrease in saliva secretion and age. 

The authors described the patients should use mouthwashes  in postoperative period. Is it means to use mouthwashes as a disinfectant?

Author Response

  1. Please provide more information for patients characteristics. The authors should describe detail for Gastrointestinal tract diseases. Such as, number of gastric surgery, esophagus surgery, colon surgery should be provide. In addition, categorized into liver, bladder and pancreas is not suitable for this study, because the quality of each operation varies widely and simple comparison/categorization of them is not possible. Please add these information in Table 1.

(Reply)

Details of the surgical site and feeding condition at 7th day after surgery were adde(d to Table 1.

  1. In Figure1 shows the large number of bacteria was detected in POD1 group. In contrast, the small number of bacteria was detected in POD7 group. Please let us know whether these groups are fasting or orally. The authors described in this article, eat orally is very important to suppress the increase in the number of bacteria in saliva of patients. The authors should add more information about fasting or orally in postoperative period.

(Reply)

Line 119-120: “Forty-seven patients were fasting on the next day after surgery, and 7 patients were fasting on the 7th day postoperatively (Table 1).” was added.

  1. In page 7, in line 170, the authors said, the number of bacteria is higher in saliva before operation in elderly patients. This could be attributed to the decrease in salivary secretion. It should need some citation literature for straightforward relationship with decrease in saliva secretion and age.

(Reply)

Reference #29 was added.

  1. The authors described the patients should use mouthwashes in postoperative period. Is it means to use mouthwashes as a disinfectant?

(Reply)

We think mouthwashes as a disinfectant.

Line 213-214: “mouthwashes” was changed to “antimicrobial mouthwashes”.

Reviewer 3 Report

The Study is interesting, However, I have a few concerns mentioned below

Abstract:

We aimed to investigate salivary bacterial counts pre-and postoperatively, and factors related to increased bacterial count postoperatively. We included 121 patients who underwent surgery under general anesthesia and perioperative oral management.” Line 15-18

Comment: Please avoid using the word “We” in the manuscript, instead you can use, “this study aimed” etc. and rephrase sentences to avoid repetition of words.

“The bacterial count in saliva was determined preoperatively and 1 and 7 days post-operatively using the dielectrophoresis and impedance measurement methods.” Line 18-19

Comment: Rephrase e.g  preoperatively, 1st and 7th days postoperatively

Increased salivary bacterial count on the day after surgery was significantly correlated with increased salivary bacterial count after oral care preoperatively” Line 25-26

Comment: Rephrase this sentence to make it clear to the reader

Keywords: perioperative oral management; bacteria in saliva; surgery; infection

Comment:  keywords need to be revised and arranged alphabetically.

Introduction:

“Postoperative complications may occur after invasive surgical procedures such as cancer surgery, cardiac surgery, and organ transplant surgery. Some complications may  be caused by pathogenic microorganisms derived from the oral cavity, and to reduce the 3risk of such complications, a clean oral environment and elimination of the infectious  source in the oral cavity are essential.” Line 34-38

Comment: Provide Reference 

 “In addition, the risk of surgical site infection (SSI) and postoperative pneumonia is significantly 4reduced by perioperative oral care during surgeries, treatment for conditions such as  esophageal cancer [1-6], gastric cancer [6], colon cancer [6-8], lung cancer [9, 10], liver cancer [11], and oral cancer [12], and procedures involving the cardiac [13] and brain [14].” Line 39-43

Comment: Please rephrase “treatment for conditions such as” with “especially during” to make the sentence continuous

“We aimed to determine patients in which the number of bacteria in saliva increases after surgery and patients and diseases for which perioperative oral care is indicated.” Line 54-56

Comment: authors may rephrase the above sentence for example, “this study was aimed to determine diseases in which perioperative oral care is indicated due to increasing salivary bacterial load after surgery”.

Materials and Methods:

Major revision of the Methodology is recommended, please mentioned the inclusion and exclusion criteria for this study, sample size calculation, and patients selected in this study who went for which surgeries. Which will help to determine diseases with more chances of postoperative SSI.

2.1. Study design and patients

This study aimed to determine factors that increase the number of bacteria in saliva on the day after surgery. The participants were patients who underwent surgery under general anesthesia at Kansai Medical University Medical Center between March 1, 202, and December 31, 2021, and who received perioperative oral care at the Department of 62 Dentistry and Oral Surgery of the Center before and 1 and 7 days after the surgery. Line 58-63

Comment: The aim mentioned here is contradicting the aim mentioned in the introduction. Here authors are saying the factors involved in increasing bacterial load after surgery. While in the introduction it seems like the author is targeting to find patients and diseases in which perioperative oral care is indicated because indications and effective methods are still not clear. Secondly, “Study design and patients” does not fit with the description of this heading, no explanation is mentioned for the study design, kindly revised it.

2.2. Variables

The following variables were examined: sex, age, primary disease, body mass index, smoking habit, drinking habit, routine dental management at a family dental clinic, serum total protein and albumin level before surgery, O’Leary plaque control record (PCR) [15], presence of periodontal pocket, number of residual teeth, denture use, oral wetness, operation time, oral intake condition on the day after surgery, and number of bacteria in saliva before and after surgery. Oral wetness was quantified according to the classification by Kakinoki et al. [16]. Saliva was collected by immersing a cotton swab in the posterior floor of the mouth. The number of bacteria in saliva was determined with a rapid bacteria quantification system (Panasonic Healthcare Co. Ltd., Osaka, Japan) using the dielectrophoresis and impedance measurement methods. Line 64-74

Comment: Details of variables should not be the part of methodology; instead, authors can mention inclusion and exclusion criteria, informed consent obtained, ethical approval,  how many patients came, who provided the perioperative care, etc.

Results:

The average logarithmic mean of number of bacteria in saliva before oral care on the  day before surgery was 5.38. After oral care before surgery, the number of bacteria in saliva decreased to approximately 1/10 of the original count. On the day after surgery, the number of bacteria in saliva increased markedly to a value of 5.68. Thereafter, the number of bacteria in saliva decreased, and the value on the 7 days after surgery was restored to that before surgery (Figure 1). Line 97-102

Comment: Please mentioned units 5.38, 5.68 and clarify the highlighted lines.

The number of bacteria in saliva decreased after preoperative oral care, increased to 105 significantly after surgery, and was restored 7 days after surgery to the number before per-106 forming preoperative oral care. Line 105-107

Comment: it is the repetition of previous statement 97-102

Discussion:

The discussion part is well written

Conclusion:

The conclusion is reflecting study's aim

References:

Please follow journal guidelines for references. DOI of each reference should be removed

Author Response

  1. Abstract: “We aimed to investigate salivary bacterial counts pre-and postoperatively, and factors related to increased bacterial count postoperatively. We included 121 patients who underwent surgery under general anesthesia and perioperative oral management.” Line 15-18. Please avoid using the word “We” in the manuscript, instead you can use, “this study aimed” etc. and rephrase sentences to avoid repetition of words.

(Reply)

Line15: “We aimed to --” was corrected to “This study aimed to --“.

  1. “The bacterial count in saliva was determined preoperatively and 1 and 7 days post-operatively using the dielectrophoresis and impedance measurement methods.” Line 18-19.

Rephrase e.g  preoperatively, 1st and 7th days postoperatively

(Reply)

Line 18-19: “1 and 7 days” was corrected to “1st and 7th days”.

  1. “Increased salivary bacterial count on the day after surgery was significantly correlated with increased salivary bacterial count after oral care preoperatively” Line 25-26

Rephrase this sentence to make it clear to the reader

(Reply)

Line25-26: “Increased salivary bacterial count on the day after surgery was significantly correlated with increased salivary bacterial count after oral care preoperatively” was revised to “Salivary bacterial count on the day after surgery was significantly increased in the patients with a large number of bacterial counts on the day before surgery after preoperative oral care”.

  1. Keywords need to be revised and arranged alphabetically.

(Reply)

Line 31, we arranged the keywords alphabetically.

  1. Introduction:

“Postoperative complications may occur after invasive surgical procedures such as cancer surgery, cardiac surgery, and organ transplant surgery. Some complications may be caused by pathogenic microorganisms derived from the oral cavity, and to reduce the risk of such complications, a clean oral environment and elimination of the infectious source in the oral cavity are essential.” Line 34-38

Provide Reference.

(Reply)

References #1-4 were added.

  1. “In addition, the risk of surgical site infection (SSI) and postoperative pneumonia is significantly reduced by perioperative oral care during surgeries, treatment for conditions such as esophageal cancer [1-6], gastric cancer [6], colon cancer [6-8], lung cancer [9, 10], liver cancer [11], and oral cancer [12], and procedures involving the cardiac [13] and brain [14].” Line 39-43

Please rephrase “treatment for conditions such as” with “especially during” to make the sentence continuous.

(Reply)

Line 41: “treatment for conditions such as --” was revised to “especially during --”.  

  1. “We aimed to determine patients in which the number of bacteria in saliva increases after surgery and patients and diseases for which perioperative oral care is indicated.” Line 54-56

Authors may rephrase the above sentence for example, “this study aimed to determine diseases in which perioperative oral care is indicated due to increasing salivary bacterial load after surgery”.

(Reply)

Line 62: “We aimed to --” was revised to “This study aimed to --“.

  1. Materials and Methods:

Major revision of the Methodology is recommended, please mentioned the inclusion and exclusion criteria for this study, sample size calculation, and patients selected in this study who went for which surgeries. Which will help to determine diseases with more chances of postoperative SSI.

(Reply)

The following sentences were added.

Line 72-74: The inclusion criteria were that the patient was at least 20 years of age and had a pre-operative panoramic X-ray examination. The exclusion criterion was patients dis-charged within 7 days of surgery.

Line 74-76: Since this study is an exploratory observational study, the sample size was not calculated statistically, and the number of patients during the study period was used as the sample size.

  1. Study design and patients

This study aimed to determine factors that increase the number of bacteria in saliva on the day after surgery. The participants were patients who underwent surgery under general anesthesia at Kansai Medical University Medical Center between March 1, 2021, and December 31, 2021, and who received perioperative oral care at the Department of 62 Dentistry and Oral Surgery of the Center before and 1 and 7 days after the surgery. Line 58-63.

The aim mentioned here is contradicting the aim mentioned in the introduction. Here authors are saying the factors involved in increasing bacterial load after surgery. While in the introduction it seems like the author is targeting to find patients and diseases in which perioperative oral care is indicated because indications and effective methods are still not clear. Secondly, “Study design and patients” does not fit with the description of this heading, no explanation is mentioned for the study design, kindly revised it.

(Reply)

Line 67: “This study aimed to determine factors that increase the number of bacteria in saliva on the day after surgery.” was deleted.

  1. Variables

The following variables were examined: sex, age, primary disease, body mass index, smoking habit, drinking habit, routine dental management at a family dental clinic, serum total protein and albumin level before surgery, O’Leary plaque control record (PCR) [15], presence of periodontal pocket, number of residual teeth, denture use, oral wetness, operation time, oral intake condition on the day after surgery, and number of bacteria in saliva before and after surgery. Oral wetness was quantified according to the classification by Kakinoki et al. [16]. Saliva was collected by immersing a cotton swab in the posterior floor of the mouth. The number of bacteria in saliva was determined with a rapid bacteria quantification system (Panasonic Healthcare Co. Ltd., Osaka, Japan) using dielectrophoresis and impedance measurement methods. Line 64-74

Details of variables should not be the part of methodology; instead, authors can mention inclusion and exclusion criteria, informed consent obtained, ethical approval, how many patients came, who provided the perioperative care, etc.

(Reply)

The descriptions were added.

Line 70-74: Inclusion and exclusion criteria, and sample size.

Line 95-96: Informed consent.

Line 98-102: Ethical approval.

  1. Results:

The average logarithmic mean of number of bacteria in saliva before oral care on the day before surgery was 5.38. After oral care before surgery, the number of bacteria in saliva decreased to approximately 1/10 of the original count. On the day after surgery, the number of bacteria in saliva increased markedly to a value of 5.68. Thereafter, the number of bacteria in saliva decreased, and the value on the 7 days after surgery was restored to that before surgery (Figure 1). Line 97-102

Comment: Please mentioned units 5.38, 5.68 and clarify the highlighted lines.

(Reply)

Line 123-124: “The average logarithmic mean of number of bacteria in saliva before oral care on the day before surgery was 5.38” was revised to “The average base 10 logarithm of the number of bacteria in saliva before oral care the day before surgery was 5.38.”.

  1. The number of bacteria in saliva decreased after preoperative oral care, increased significantly after surgery, and was restored 7 days after surgery to the number before performing preoperative oral care. Line 105-107

It is the repetition of previous statement 97-102

(Reply)

Line 132: “The number of bacteria in saliva decreased after preoperative oral care, increased significantly after surgery, and was restored 7 days after surgery to number before performing preoperative oral care” was deleted.

”.

  1. References:

Please follow journal guidelines for references. DOI of each reference should be removed.

(Reply)

DOI was removed.

Round 2

Reviewer 3 Report

Dear Authors,

I am satisfied to see the improvement and answer reviewer comments.